# Molecular Dynamics Simulations of Curved Lipid Membranes

**DOI:** 10.3390/ijms23158098

**Published:** 2022-07-22

**Authors:** Andreas Haahr Larsen

**Affiliations:** Department of Neuroscience, University of Copenhagen, 2200 Copenhagen, Denmark; andreas.larsen@sund.ku.dk

**Keywords:** molecular dynamics (MD), lipid membrane, membrane curvature, vesicle, free energy of binding

## Abstract

Eukaryotic cells contain membranes with various curvatures, from the near-plane plasma membrane to the highly curved membranes of organelles, vesicles, and membrane protrusions. These curvatures are generated and sustained by curvature-inducing proteins, peptides, and lipids, and describing these mechanisms is an important scientific challenge. In addition to that, some molecules can sense membrane curvature and thereby be trafficked to specific locations. The description of curvature sensing is another fundamental challenge. Curved lipid membranes and their interplay with membrane-associated proteins can be investigated with molecular dynamics (MD) simulations. Various methods for simulating curved membranes with MD are discussed here, including tools for setting up simulation of vesicles and methods for sustaining membrane curvature. The latter are divided into methods that exploit scaffolding virtual beads, methods that use curvature-inducing molecules, and methods applying virtual forces. The variety of simulation tools allow researcher to closely match the conditions of experimental studies of membrane curvatures.

## 1. Introduction

Membrane curvature plays a central role in cellular trafficking. This role is apparent in the formation of vesicles during endo- and exocytosis. Membrane curvature also plays a more direct role, since differences in curvature can target curvature-sensing lipids, peptides, and proteins to specific locations [1]. However, by which mechanisms does membrane curvature induce the sorting of these molecules? We know that there is a broad variation of membrane curvatures in mammalian cells: from the plasma membrane, which is flat at the nanoscopic scale, to the highly curved surfaces of the endoplasmic reticulum, the Golgi apparatus, membrane protrusions [2], or organelles, such as endosomes and lysosomes (Figure 1). However, how is membrane curvature generated and maintained in the different cellular compartments? 

Molecular dynamics (MD) simulations provide molecular insight with high spatial and temporal resolution and MD is thus a strong complementary tool to experimental studies when addressing these questions. As we shall see, there are various methods for simulating curved lipid membranes with MD and, thanks to dedicated scientists, many of these are implemented, documented, and made available as open-source software. However, selecting the best method for a given problem is not trivial. One strategy is to model whole organelles [3,4]. Albeit attractive and promising, only short simulations have so far been made for whole organelles [5]. In this review, the focus is therefore on smaller model systems of curved membranes, with a more moderate computational cost, and the questions that may be unraveled by each model system are discussed. I will not provide details about the levels of molecular resolution in different lipid models, as this has already been covered [6]. The review is also limited to biological molecules, although nanotubes and nanoparticles can also induce and sense membrane curvatures [7,8], which is relevant, e.g., in bioengineering. 

The motivation for writing the review was thus to provide an overview of available tools for researchers who are interested in applying MD simulations to study the interplay between membrane curvature and proteins, peptides, and various lipid types. With the review in hand, it should be easier to find the best-suited simulation approach for a given scientific question. 

## 2. Biological Background

A variety of proteins and lipids induce membrane curvature or are sorted by membrane curvature. One group is the coating proteins COP I and COP II that mediate budding and vesicle transport between the endoplasmic reticulum and the Golgi apparatus [9], and the functionally related protein, clathrin, which also facilitates vesicle formation by coating [10] (Figure 1). The superfamily of BAR domain-containing proteins is another notable example. BAR domains form dimers whose shape resembles that of a banana or a crescent moon. They can scaffold the membrane into a curvature, defined by this shape, when nucleating on the membrane surface [11]. BAR domain-containing proteins can also create membrane curvatures by inserting amphipathic helices in a wedge-like mechanism, or by mere crowding [12,13,14]. BAR domains are involved in several membrane-reconstructing processes, including endocytosis and the formation of T-tubules, and they are thus key players in membrane remodeling. Another interesting curvature-dependent group is antimicrobial peptides. These peptides can penetrate and form pores in bacterial membranes, which make them a potential drug candidate that would constitute a much-needed alternative to conventional antibiotics [15]. Fusion peptides are another group of peptides that can penetrate and remodel lipid membranes. These peptides are part of fusion proteins, which are involved in viral entry and therefore constitute a potential target for anti-viral drugs [16]. Lipids themselves can also generate and sustain curvature. This can be achieved through lipid clustering and the formation of lipid rafts [17], or simply by an asymmetric lipid distribution in the inner and outer leaflets [18]. These are just a few examples that illustrate the abundance of membrane-curvature-inducing and curvature-sensing molecules. Other examples will be mentioned in the text, as the different methods for simulating membrane curvature are discussed. 

## 3. Spontaneously Induced Membrane Curvature

In this section, I will give an overview of methods that simulate how membrane curvature can be spontaneously induced by proteins, peptides, or lipids (i.e., without applying any virtual forces or scaffolding dummy beads). The starting configuration is either randomly distributed lipids or a flat membrane, which is then reshaped into a curved membrane. 

### 3.1. Self-Assembly

The least biased way to simulate spontaneous curvature is to allow the lipids (and proteins) to self-assemble from a randomly distributed initial configuration. This approach was used by Cooke and Deserno to investigate the role of the lipid packing parameter on membrane shape [19]. As simulated self-assembly is computationally expensive, the authors used an approximate force field with just three beads per lipid (i.e., around 50 atoms per bead) and implicit water. That is, the effect of water was introduced by an attractive force between beads representing the hydrophobic lipid tails. 

Other methods, which will be discussed below, are computationally less expensive, enabling the use of force fields with higher resolution and explicit water. That level of resolution is necessary if molecular details rather than large-scale membrane changes are of interest. These methods use a preassembled lipid bilayer as the starting configuration. 

### 3.2. Bending of Plane Bilayers with Periodic Boundary Conditions

A straight-forward strategy for monitoring spontaneously induced curvature is to simulate a flat membrane with periodic boundary conditions and probe the effect of adding curvature-inducing molecules, such as peripheral membrane proteins. This method has been applied in studies of, e.g., F-BAR domains [20] and N-BAR domains [21]. Some integral membrane proteins also induce spontaneous local curvature to obtain a better match between its transmembrane part and the hydrophobic core of a lipid bilayer [22]. The non-structural protein 4A of dengue virus is an example of a transmembrane protein that induces membrane curvature as a first step towards vesicle formation [23]. Intriguingly, MD simulations of a flat membrane have also been used to show that membrane curvature can be induced by mere protein crowding, i.e., by high concentrations of proteins, which have little or no affinity for lipid membranes by themselves [24] (Figure 2A).

A challenge for membrane curvature in planar membranes is the repulsive force from the periodic boundary condition, which inevitably leads to underestimation of the induced curvature. This is particularly critical for large perturbations of the curvature. The repulsive effect can, to some extent, be circumvented by simulating a larger membrane patch such that edge effects are negligible. However, smaller membranes are attractive due to lower computational costs, so methods have been developed to overcome the challenge of combining periodic boundary conditions with membrane curvature without making excessively large systems, as described in the following. 

### 3.3. Bicelles and Bicelle-to-Vesicle Transitions

Small bilayer patches in solution are energetically unstable due to the exposed hydrophobic lipid tails at the rim. The bilayer patch can, however, be solubilized by detergents or short-tailed lipids to form bicelles, or by amphipathic helices or polymers to form nanodisc-like particles [27]. Bicelles are the most interesting of these systems in the context of membrane bending, as the flexible rim of detergents or short-tailed lipids allow the membrane to bend. Moreover, lipids can translocate from one leaflet to the other via the rim to equilibrate the density difference that inevitably builds up upon bending. Importantly, this pressure difference cannot be equilibrated in flat, periodic membranes, unless an artificial pore is formed and sustained. In that case, the artificial pore can be used by the lipids to translocate from the dense (concave) to the less dense (convex) leaflet. One example of the use of bicelles for studying membrane curvature is an MD-based investigation of curvature-inducing amyloid-β peptides [28]. In that study, the peptides bound to a metastable bicelle, allowing the bicelle to close in on itself and form a vesicle. In absence of the amyloid-β peptides, the energy barrier between bicelle and vesicle was too high for a spontaneous transformation to happen, but the peptides lowered the energy barrier. A similar approach was used to investigate the curvature-inducing properties of the reticulum homology domain (RHD) of ER-phagy receptor FAM134B [25] (Figure 2B). The clustering of protein on the bicelle led to a bicelle-to-vesicle transition. Simulations of bicelle-to-vesicle transitions have also been used to investigate the curvature induction of protein complexes from dengue virus [29].

### 3.4. Budding of Plane Membranes

The curvature-inducing properties of FAM134B-RHD were, in a later study, investigated by a different approach, using a plane membrane with a periodic boundary condition as the initial configuration [26]. The membranes had an imbalance between the number of lipids in the upper and lower leaflet, and was thereby in a metastable initial state, with a budded membrane constituting a more stable state. That is similar to the bicelle-to-vesicle transition which was also in a metastable state, but in the case of the bicelles, the initial plane membrane was instead unstable due to the exposed hydrophobic edges. FAM134B-RHD induced budding of the membrane by lowering the energy barrier between the plane membrane and the energetically favorable budded membrane state (Figure 2C). The budding approach allows for protrusions of different sizes [26], in contrast to the bicelle-to-vesicle transition method, where the final vesicle diameter is constrained by the number of lipids in the bicelle. Thus, the budding approach can be used to determine the preferred curvature of different combinations of lipids and of curvature-inducing proteins. 

### 3.5. Lipid Bilayers with Free Edges in One Direction

The last method, which is included in this section, is a modification of the simple plane membrane. Mahmood et al. investigated the curvature-sensing properties of the F-BAR domain Pacsin1 by generating a bilayer with free edges in one direction, and no pressure coupling in the membrane plane [30]. Thus, the membrane could bend in one direction without having to overcome a restoring force from the periodic boundary condition. A regular grid of Pacsin1 was placed on the membrane and spontaneously induced curvature was monitored. The system was simulated for 2 μs, meaning that the system was surprisingly stable despite having the hydrophobic tails exposed to water in one direction. The same method has been used to investigate the membrane-bending properties of the endosomal sorting complex ESCRT [31].

## 4. Vesicle Formation

Vesicles are abundant in all cells and, depending on their size, they consist of moderately to highly curved lipid membranes. Formation and fusion of vesicles are vital in many biological processes, from cellular trafficking and synaptic signaling to protein degradation. Computational studies employing MD have been used to investigate, e.g., how viral fusion is mediated by the fusion peptide hemagglutinin [32], how lipid composition affects vesicle fusion [33], and how vesicle size and membrane thickness is related [34]. A prerequisite for such studies is the formation of vesicles in silico, and I describe methods for solving that task in the following section. 

### 4.1. Self-Assembled Vesicles

Vesicles can be formed by allowing the phospholipids to self-assemble. This has the obvious advantage that self-assembly itself can be investigated, e.g., to probe the propensity of different lipids towards vesicle formation [35] (Figure 3A). However, there are more effective ways to form vesicles if the self-assembly process is not the point of interest. 

### 4.2. CHARMM-GUI Vesicle Builders

The CHARMM-GUI [38] includes two tools for building vesicles: the first is part of the Martini Maker [36], the other is part of the PACE CG builder [39]. The PACE CG supports both the coarse-grained Martini force field [40], with about 4 heavy atoms per bead, and the united atom PACE force field [41], with one bead per heavy atom. The dedicated Martini Maker is more widely used for vesicle generation (as judged by inspecting the papers that cite the respective vesicle builders), possibly because it includes more Martini lipids, and includes a protocol for dealing with the difference in lipid density between the inner and outer leaflet of the vesicle. The latter is facilitated by artificial pores in the vesicle, which allow translocation of lipids between leaflets. The pores are preserved by positional restraints, which are released after an initial equilibration, resulting in an unbroken and equilibrated vesicle for further studies (Figure 3B). 

### 4.3. From Triangular Surfaces to Lipid Vesicles 

Vesicles can also be formed by use of triangular surfaces. Triangular surfaces consist of a grid of nodes which represents the corners of triangles that can span any surface. They are used in computer graphics to approximate basically any shape, from spheres to cartoon characters. The triangulated surfaces can be generated by third-party software, which is not made specifically for MD simulations. After generating the shape, the next step towards running an MD vesicle simulation is to fill the triangulated surface with lipids. This step has been implemented in LipidWrapper [42] and TS2CG [5]. LipidWrapper uses a “cookie cutter” method to fill each triangle with lipid patches cut out from an equilibrated flat membrane made by the user. TS2CG, on the other hand, first generates a finer grid of nodes from the original triangulated surface, then calculates the normal to the curved surface, and places lipids at the new nodes oriented along the normal (Figure 3C). Both methods provide well-equilibrated vesicles, and are well-documented. They also have the flexibility to form various other shapes. Just like the CHARMM-GUI Martini Maker’s vesicle builder, TS2CG uses pores to equilibrate the difference in lipid density between the inner and outer leaflets. TS2CG also has a built-in tool to generate predefined shapes without providing a triangular surface as input [43].

Importantly, both triangular surface-based methods as well as both CHARMM-GUI vesicle builders can generate vesicles with embedded membrane proteins (Figure 3E). 

### 4.4. Meshless Grid

The meshless grid model is an alternative to the triangulated surfaces [44]. In this model, beads resembling the nodes in the triangulated surface represent the lipids, but the beads are not connected by bonds. This model, which also uses implicit water, includes parameters for tuning the membrane curvature, and has been used to study the large-scale formation of curved membranes, e.g. as a result of BAR domain binding [45]. However, methods for converting the meshless model to a higher molecular detailed representation are currently unavailable, limiting its application. 

### 4.5. Transformation from Flat Membranes to Vesicles

Vesicles can also be formed by bending a flat membrane into a predefined vesicle shape, as implemented in the program BUMPy [37]. A key feature of BUMPy is the ability to estimate the asymmetry of the inner and outer leaflets and adjust the density of the leaflets in the flat membrane before transforming it into a vesicle. Thus, no artificial pore is necessary for equilibration of lipid density differences between leaflets. This method is versatile, like the triangular surface-based methods, as BUMPy can form differently shaped membranes, including tori, hollow cylinders, or bilayers connected by a cylindrical pore (Figure 3D).

## 5. Non-Spontaneous Membrane Curvature Sustained by Scaffolding Beads or Virtual Forces

Protein or lipids can induce membrane curvature, but membrane curvature can also sort molecules to a specific location. To investigate curvature-dependent sorting, it is useful to be able to sustain a prefixed membrane curvature in an MD simulation. Some of the methods described above can generate starting configurations for curved membranes (TS2CG, LipidWrapper, and BUMPy), but the initial shape is not necessarily sustained throughout the simulation. One way to sustain curvature is by generating vesicles or lipid tubules, whose curvatures are defined by their radii. However, smaller systems than whole vesicles or tubules are often preferable due to computational costs and, in addition to that, it is beneficial for sorting studies to be able to simulate membranes with multiple curvatures so that movement from one curvature to another can be monitored. Luckily, a range of methods have been developed in recent years to meet these needs. They can be divided into three overall groups, where the curvature is sustained by either virtual beads, curvature-inducing molecules, or virtual forces.

### 5.1. Virtual Beads and Solid Supports

One way to sustain a curved membrane is by using beads that are fixed in space. By imposing attractive or repulsive forces between beads and lipids, the membrane adapts to the curvature dictated by these beads. This can be achieved by forming walls of virtual beads above and below a curved membrane and imposing repulsive forces between the beads and the lipid tails of the membrane (Figure 4A). This method was introduced by Yesylevskyy et al. [46], who used the method to study the permeability of curved membranes [47]. The method is also implemented as part of TS2CG (from version 1.1) [43]. This wall-of-beads method is very effective in sustaining a fixed curvature and is straightforward to set up using TS2CG [43]. However, the method has some core limitations [48]: First, the walls have to be fixed before equilibration, so the membrane height must be known a priori. Second, for mixed membranes, the restraining caused by the walls may be suboptimal for some lipids with tail groups that are shorter or longer than the average, and thus are non-ideal for simulating lipid clustering. Third, the membrane may need to expand vertically during equilibration, if the area per lipid headgroup is not perfectly chosen a priori, or the membrane is perturbed, e.g., by peptide insertions. In that case, the membrane exceeds beyond the barriers imposed by the wall restraint, so edge effects must be considered when analyzing the data. Another point to be aware of is the difference in lipid density between concave and convex leaflets. This imbalance can be equalized by using bicelles [46,47], where the density difference can be relieved through the rim. In periodic membranes, on the other hand, the lipids are restrained to one leaflet, so each leaflet must contain an equal amount of concave, flat, and convex patches to avoid density imbalance. Albeit having some limitations, the wall-of-beads method is a valuable tool, as membrane curvature can be tuned and controlled. Moreover, simulations with several curvatures in one system can be generated.

An alternative method, also using virtual beads, is to place the membrane on a solid support of beads (Figure 4B). This surface can contain either stalks [49] or holes [50] and attractive forces ensure that the membrane adapts to this surface. The curvature can, to some extent, be controlled by tuning the attractive force and the size of the holes or distance between stalks. This method was first introduced to investigate the effect of solid supports on membranes in in vitro experiments [49,50], but has later been applied to study the sorting of curvature-sensitive epsin N-terminal domain (ENTH) [51] and various ANTH domains [52]. As opposed to the wall-of-beads method, the membrane is only restrained from one side, so the membrane height needs not to be known a priori, and the method can therefore better accommodate lipids with different tail lengths. Providing a small water layer between the solid support and the membrane ensures that the disturbance of the membrane is minimal [49], which is in line with experimental findings [53]. The more relaxed membrane does, however, come with a price: it is more tedious than in the wall-of-beads approach, to set up the system and vary the curvature, and the solid support method has only been used for large systems (so far), which increases the computational cost. 

### 5.2. Curvature-Inducing Molecules

Lipids are known to induce various curvatures, largely due to their packing parameter [19]. Although it is debated how large the effect of the lipid’s packing parameter really is [57], the difference in lipid packing parameters has been exploited in MD simulations to generate curved membranes by clustering lipids with positive and negative packing parameters in the opposing leaflet (Figure 4C). Lipid asymmetry-induced curved membranes were used to study, e.g., the mechanosensitive protein MscL [54] and to probe how cholesterol affects membrane shape [58]. There is a strong enthalpic pressure towards lipid mixing between clusters in the same leaflet, so to sustain the initial curvature, repulsive forces between lipids in the different clusters must be applied. To suppress membrane fluctuations and sustain a given curvature, position constraints can also be applied. Albeit elegant, the method has some limitations. Obviously, the lipid composition is limited, as specific lipids are utilized to create the curvature, and the composition may not match the physiological conditions. It is, however, possible to maintain the curvature by only having anisotropy in the lower leaflet, leaving the upper leaflet free for variations in the lipid composition. Another limitation is that the curvature cannot easily be tuned. A third limitation is that one has to define new bead types to implement the repulsive forces that suppress lipid mixing (see example of implementation at https://github.com/andreashlarsen/Larsen2022-review), making the setup elaborate. Moreover, the repulsive forces that prevent lipid mixing also prevent the release of pressure, which is built up by bending. Normally, this pressure can be released by lipid diffusion from concave to convex patches but the repulsive forces hinder this.

It may be possible to use other curvature-inducing molecules to obtain a desired curvature, e.g., by allowing BAR domains to bind to one membrane leaflet and probe the curvature sensing of other peripheral proteins using the opposing leaflet. One could alternatively simulate peptidoglycans, which scaffold the bacterial surface [59] and thus sustain a given curvature. None of these approaches have, however, been attempted (to the author’s knowledge), probably because more adaptable methods exist, as will be covered in the next section. These methods are more adaptable in the sense that lipid composition and membrane curvature can easily be varied and controlled.

### 5.3. Virtual Forces

Virtual forces constitute an alternative to virtual beads, and the forces can be applied in various ways depending on the scope of the study. One method is to pull a tether from a flat membrane by applying a pulling force to a patch of lipids [60,61] (Figure 5D). This method has been used to investigate lipid sorting in mixed, asymmetric membranes [62]. Membrane tethers contain a high variety of curvatures, from highly concave in the interface between the flat membrane and the tether, to highly convex at the top of the tether. This makes it an attractive method for curvature-sorting studies. The curvature cannot be controlled directly but is determined by the lipid composition and amplitude of the pulling force. 

A more controlled way to induce and sustain curvature by virtual forces, is to impose a specific radius by a collective variable. A couple of such curvature-controlling collective variables have been derived in recent years [63,64,65]. An elegant implementation, EnCurv, exploits PLUMED [66,67] to implement a collective variable that can enforce a specific curvature to a bicelle or a patch of a periodic membrane [48] (Figure 4D). A notable attribute is that the PLUMED implementation makes the method agnostic to force field or MD engine and easy to set up. Moreover, the bilayer is less constrained than in the wall-of-beads method, albeit the curvature is still very controlled and tunable. When simulating bicelles, the leaflet lipid density gradient can be relieved by exchange via the rim. For periodic membranes, on the other hand, temporary artificial pores must be formed during equilibration for lipid exchange between leaflets, as discussed previously for the vesicle builder in the CHARMM-GUI Martini Maker (Figure 2B). EnCurv and related methods sustain a uniform curvature over the affected lipid patch or bicelle, which can be very useful. This is, however, also a limitation of the method, as the lack of multiple curvatures means that several parallel simulations may be needed if curvature sorting is the scope of the study.

The last method that uses virtual forces is membrane buckling. This method starts with a plane membrane, and by decreasing the box size in the membrane plane, a lateral stress is built up, which is released by spontaneous transformation from a flat to a curved membrane (buckling). The decreased box size can either be obtained by increasing the pressure in one direction [56,68,69] or by applying a force in that direction [70]. Alternatively, the box size can be decreased by scaling all particle coordinates along one direction [55,71,72,73], followed by equilibration with the box size fixed. In the latter approach, the compression (degree of buckling) can be directly controlled. By fixing the box size after the buckling occurs, a lateral force effectively acts on the membrane. This reactive force prevents the membrane from reshaping into a flat geometry, so the curvature is approximately sustained (Figure 4E). To prevent membrane fluctuations and to stabilize a given membrane curvature, position restraints can be applied to one membrane leaflet, thus sustaining curvature while keeping the lipids in the opposing leaflet relatively unperturbed [56]. Just like the membrane tethering method, the curvature of the buckled membrane is not directly controlled, but indirectly given by the lipid composition and the degree of compression. Membrane buckling has been used to study, e.g., the curvature sensing of amphipathic helices from antimicrobial peptides [71], from viral scission protein [55], and from α-synuclein [56]. It has also been used to study the curvature sensing of cardiolipin [68,72], and lipid sorting of various lipid types in mixed bilayers [69].

### 5.4. Free Energy Calculations

The ability to generate and sustain curved membranes in MD provides the basis for free energy calculations. Energetics are useful for quantifying bending properties of the membrane and collective variables have been applied to calculate bending moduli of different lipid bilayers [63,64,65]. The affinity of peptides, proteins, and lipids towards specific curvatures can also be quantified with free energy calculations [74]. Recently, Stroh and Risselada described how membrane buckling and umbrella sampling can be used to calculate the relative free energy of the binding of curvature-sensing peptides ALPS and α-synuclein as a function of membrane curvature [56].

## 6. Comparison with Experiments

Membrane curvature can be generated and measured experimentally, and comparison with experimental results are essential for verification of MD simulations. The most common experimental techniques for measuring curved membranes are electron microscopy and fluorescence microscopy. Electron microscopy can directly probe large-scale structural changes of the membranes. One example is a study of the transitions from vesicles to tubules induced by the BAR domain of amphyphysin [75] (Figure 5A). In fluorescence microscopy, on the other hand, a bulk lipid is fluorescently marked. Importantly, this lipid must be homogeneously distributed in the membrane, i.e., not be curvature-sensitive itself. The proteins or lipids of interest must likewise be fluorescently labeled. The relative density of the proteins or lipids of interest is quantified by the ratio of fluorescence intensities, I_bulk-lipid_/I_molecule-of-interest_, at membrane segments with different curvatures.

By these means, curvature and sorting can be monitored. Analogous to the simulation methods described above, the obvious next challenge is, then, how to generate lipid membranes with varying curvatures. The various experimental methods are described in more detail elsewhere [1,76,77] but this section is included to provide a more direct comparison to the presented simulation tools. Evidently, virtual forces or virtual beads cannot be exploited in experimental methods, but external (non-biological) forces can have very similar effects. 

### 6.1. Vesicles

By using an experimental setup with vesicles of varying diameter, the curvature can be linked to the fluorescence intensity, thus quantifying curvature-induced sorting [78,80] (Figure 5B). Flow cytometry is a promising alternative to microscopy for high-throughput experiments (thousands of vesicles) [81]. By comparison with MD simulations of vesicles with the same lipid composition and protein content, researchers can investigate molecular details that are hard to probe experimentally. One example is a study wherein the authors combined electron microscopy with coarse-grained MD to study how α-synuclein deforms vesicles [82]. The MD simulations revealed how α-synuclein is located on the most convex area of the deformed vesicles. In another study, vesicle fusion was simulated with MD and the results agreed qualitatively with experimentally determined fusion rates. The simulations provided a better understanding of how the lipid composition affected vesicle fusion [33]. 

### 6.2. GUVs with Tubules

GUVs with diameters in the order of 1 um are flat at the nanoscopic scale. Including curvature-inducing proteins or lipids in the GUV may result in spontaneous tubulation on the GUV surface. As the tubules have high membrane curvature, they constitute a target location for molecules that prefer to be located at non-zero curvature.

The membrane tubules can also be generated in an optical tweezer experiment. To this mean, biotin-labeled lipids are included in GUV membranes. A bead coated with streptavidin, which has high affinity for biotin, is positionally restrained by optical tweezers and, using the bead, a tubule can be pulled from the GUV (Figure 5C) [79,83]. With this method, curvature can be tuned by the pulling force of the optical tweezer. Interestingly, this method is very analogous to the MD approach, where a tubule is pulled from a flat membrane using virtual forces [60] (Figure 5D). 

### 6.3. Solid Support

Solid supports can be simulated with MD [49,50], as discussed in the previous section. These in silico setups are directly inspired by in vitro equivalents [84,85,86]. Intriguingly, this makes it possible to directly compare MD with experiments. Unfortunately, direct comparative studies (with the same sample being investigated by solid supports in silico and in vitro) have not, to my knowledge, been conducted yet.

### 6.4. Example: Combining MD and Experiments for a Curved Membrane

A great example of how experiments and simulations complement each other is a study of the effect of polyunsaturated lipids on endocytosis [60]. Using electron microscopy, Pinot et al. showed that polyunsaturated membranes are more prone to membrane fission than monounsaturated membranes. Coarse-grained MD simulations were performed, as the lipid composition could be monitored and controlled better than in the experimental setup. In the simulations, a tubule was pulled from either mono- or polyunsaturated membranes, and it was observed how less pulling force was needed to cause a fission event for the polyunsaturated membranes (Figure 5D). The simulated trajectories provided molecular insight into how polyunsaturated lipids induce packing defects and better adapt to new curvatures due to their flexible tails. 

## 7. Challenges and Future Directions

There has been a rapid and promising development of methods for simulating membrane curvature in recent years, which allows for some interesting future directions, which will be discussed in the following section, along with some of the challenges that remain.

One practical limitation is that many of the simulation methods are only implemented within a limited number of force fields, water models, and MD engines. Implementation via PLUMED is possible for some methods, in particular those based on collective variables [48], which make these methods immediately available for most users. Other methods are force-field-specific. This challenge is general for many MD methods and as such is not limited to simulations of curved membranes but it is nonetheless an important challenge to face, also in this context. 

Another challenge is of more political than scientific nature and concerns the expansion and development of scientific programs. Novelty is, for good reasons, credited in science when publishing and getting grants. Thus, the scientific “reward” for maintaining and refining existing programs is limited, although these are pivotal for the scientific community. Given the necessary resources, existing tools could be expanded to accommodate more force fields, more lipid types, and more geometries. For example, it would be useful with collective variables for periodic functions, similar to the Fourier shapes available in TS2CG [43].

An area that has experienced rapid development is the transition between different realms of resolution. It is pivotal to choose the right level of detail for a given simulation to balance accuracy and computational cost [6]. Some parts of the system may require a higher level of detail than others, e.g., binding sites vs. bulk water, so a dual-resolution approach combining coarse-grained and atomistic detail is therefore alluring [87], which is analogous to QM/MM simulations [88]. Proof-of-principle studies have shown that this is possible, despite the challenges of combining different force fields. In a study by Orsi et al., antimicrobial peptides in atomistic resolution bound to coarse-grained lipid membranes were simulated [89], and in a more recent study by Liu et al., a vesicle with atomistic interior and coarse-grained exterior was simulated [90]. This is indeed a promising approach. A simpler approach is to first simulate with one resolution, then convert the system (or a subset of it) to another resolution and continue the simulation, e.g., to validate a coarse-grained binding pose with all-atom MD [91]. The conversion from lower to higher resolution is challenging, as information has to be added to solve this inverse problem. Fortunately, several programs that deal with this problem exist: LipidWrapper [42] and TS2CG [5] can convert triangular surfaces to molecular detail, whereas CG2AT [92] and the backward script [93] convert from coarse-grained to atomistic resolution. 

Finally, it is not sufficient to be able to simulate curved membranes; the data should also be analyzed. Due to the non-standard membrane geometry, tools developed for plane membranes, such as LiPyphilic [94], are often inadequate. More specialized methods have, however, been developed [58], including the package Memsurfer, which provides a range of analytical methods designed for curved membranes [95], and CurD for analyzing diffusion rates in curved membranes [96].

The number of studies that directly combine MD simulations and experiments for curved membranes (e.g., Ref. [60]) are still limited. However, with the tools reviewed here, it is feasible for a broad scientific community to set up and run simulations that match the given experimental conditions closely, providing invaluable molecular insight. 

## 8. Conclusions

Investigating curvature sensing and curvature induction in MD simulations is a daunting task, as the researcher must consider several factors, including how to generate and sustain out-of-equilibrium membrane geometries; how to perturb the membrane as little as possible, despite this non-plane geometry; and how to balance computational costs against having a sufficiently large system, which include the relevant curvature(s) and negligible edge effects. Fortunately, and thanks to dedicated researchers and software developers, a powerful collection of computational tools and protocols have been developed in recent years. This includes methods for generating the starting configurations for membranes of arbitrary shapes, methods for sustaining these shapes if needed, methods for shifting between different levels of molecular detail, and methods for analyzing the simulated data. These efforts, along with the parallel development of experimental methods, are pivotal for unraveling the role of membrane curvature in the diseased and the healthy body.

## Figures and Tables

**Figure 1 ijms-23-08098-f001:**
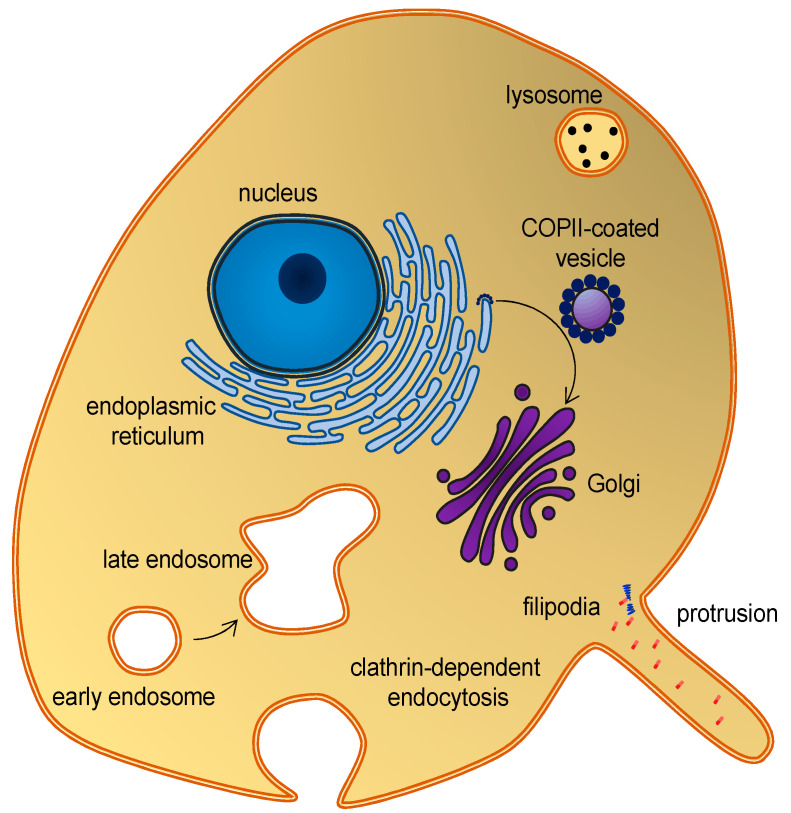
Eukaryotic cell and organelles, with examples of various membrane curvatures. Some curvature-inducing proteins are also shown, see the main text.

**Figure 2 ijms-23-08098-f002:**
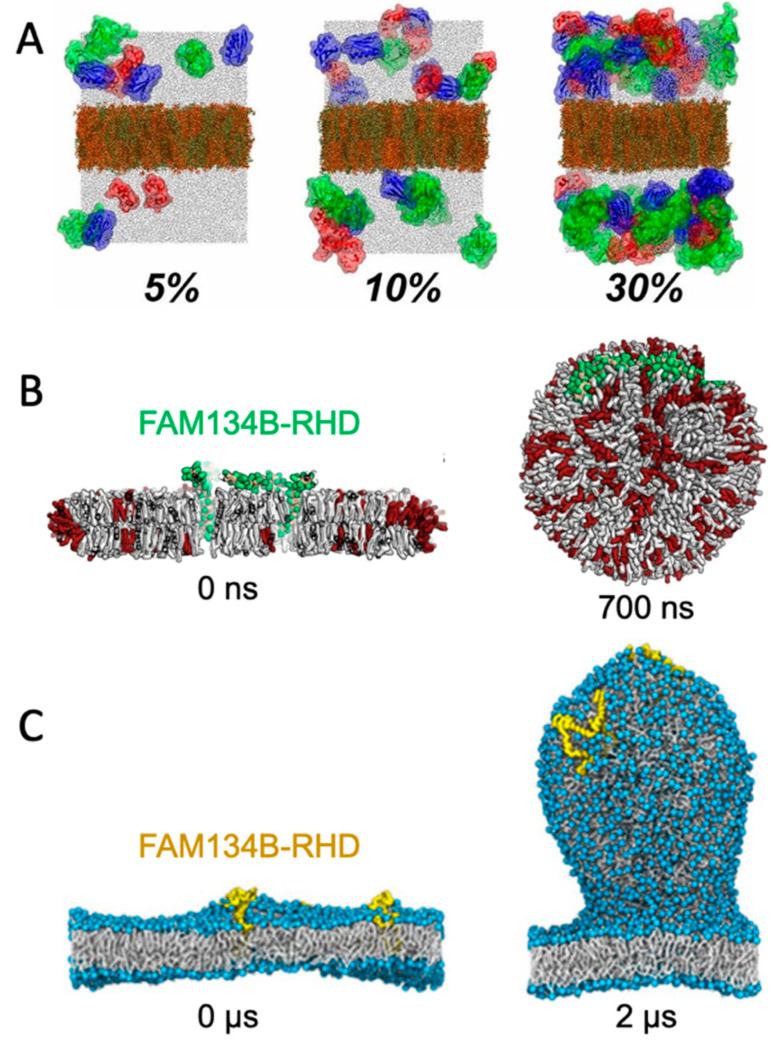
Simulating membrane curvature induction. (**A**) Bending of plane periodic bilayer by protein crowding. Figure adapted with permission from Ref. [24] (copyright 2019, PNAS license). (**B**) Bicelle to vesicle transition, adapted from Ref. [25]. (**C**) Tethering from a plane membrane, adapted from Ref. [26]. If not stated otherwise, the original figures are published under a creative common license.

**Figure 3 ijms-23-08098-f003:**
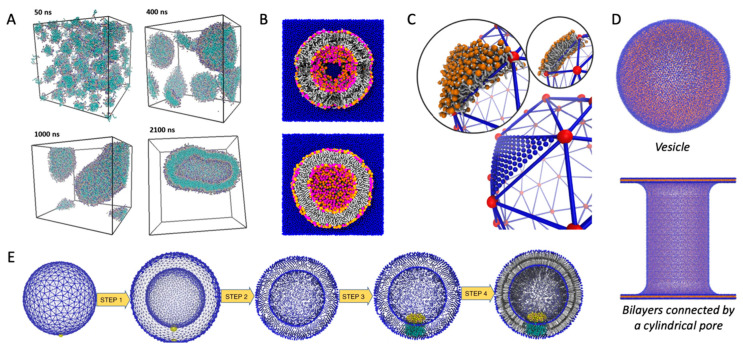
Methods for vesicle generation. (**A**) Self-assembly of vesicle, adapted from Ref. [35]. (**B**) CHARMM-GUI Martini Maker, adapted from Ref. [36]. Top shows the cross-section of a vesicle with pores for lipid translocation between leaflets, bottom shows the vesicle after pore-closure. (**C**) Generation of vesicles by triangular surfaces, using TS2CG. Adapted with permission from Ref. [5] (copyright 2015, ACS Publications). (**D**) Curvature generated from transition of a flat membrane, using BUMPy. Adapted from Ref. [37]. (**E**) Insertion of membrane protein in a vesicle, using TS2CG. Adapted from Ref. [5]. If not stated otherwise, the original figures are published under a creative common license.

**Figure 4 ijms-23-08098-f004:**
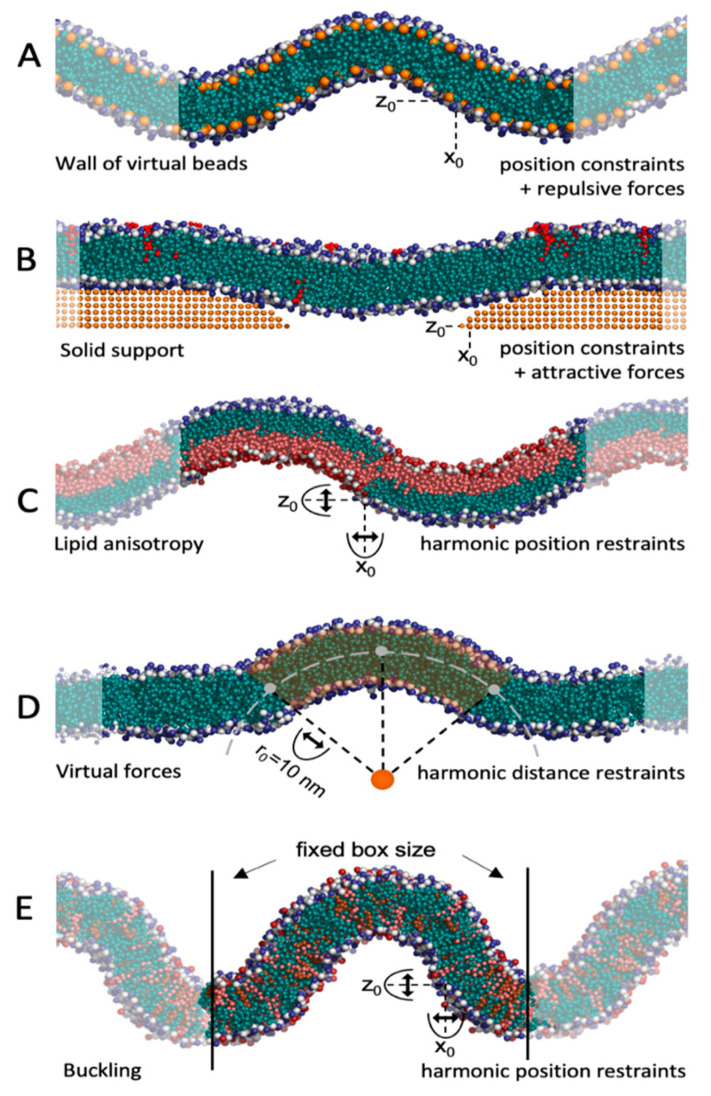
Methods for setting up and maintaining curvature. (**A**) Scaffolding approach, employing a ‘wall’ of virtual beads (orange) that sustain membrane curvature by repulsive forces between the beads and lipid tails. Pure POPC membrane, set-up using TS2CGv1.1 [43]. The semi-transparent membrane is the periodic extension of the simulated box. (**B**) Scaffolding approach, where the membrane is attracted to a solid support with a hole. A cross-section through the hole is shown. Membrane of DOPC (blue) and POPI_P2_ (red). Set-up using the protocol in Ref. [50]. (**C**) Lipid asymmetry-induced curvature. The anisotropy is maintained by virtual repulsive forces between POPC (blue) and POPS (red). Membrane fluctuations are suppressed by harmonic position restraints. Set up following Ref. [54]. (**D**) Virtual forces can generate and maintain a membrane curvature defined by a set of collective variables. Here, the virtual forces are harmonic restraints on the distance between a virtual bead (orange) and a section (yellow) of a POPC membrane. Set-up using EnCurv [48]. (**E**) The membrane is compressed in one direction (buckling) and then the box dimensions are fixed. Membrane fluctuations are avoided by position restraints. Mixed membrane of POPC (blue), POPS (red) and cholesterol (brown). Set up using Refs. [55,56]. (All original figures, generated for this review).

**Figure 5 ijms-23-08098-f005:**
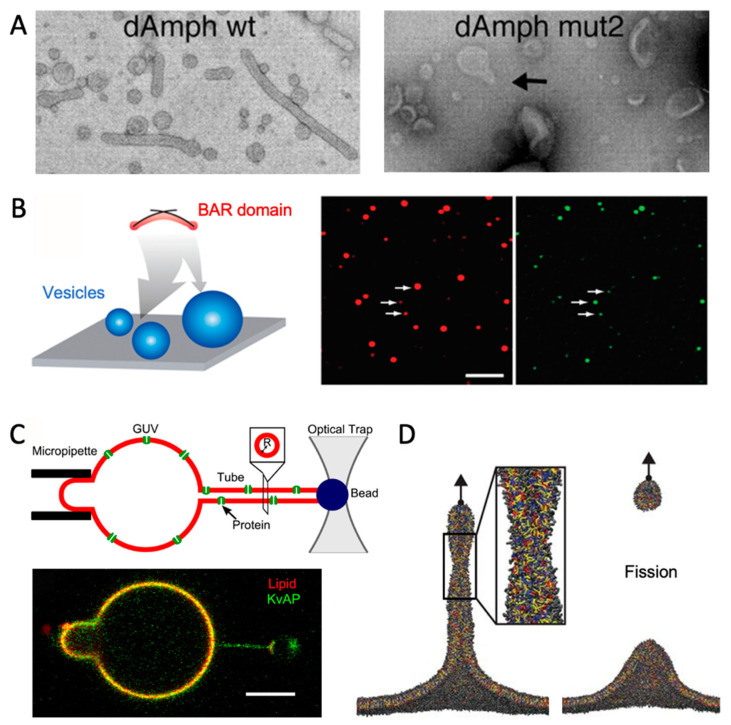
Experiments for measuring membrane curvature. (**A**) Electron microscopy images of how amphyphysin (wild type and mutant “mut2”) induce transitions from vesicles towards tubules. Adapted with permission from Ref. [75] (copyright 2004, The American Association for the Advancement of Science). (**B**) Fluorescent liposomes (red, middle panel) on a surface with bound BAR domains (green, right panel). Adapted with permission from Ref. [78] (copyright 2009, John Wiley and Sons). (**C**) Tethering from GUV with optical tweezer, adapted with permission from Ref. [79] (copyright 2014, Elsevier). (**D**) Simulated tethering and fission event, adapted with permission from Ref. [60] (copyright 2014, The American Association for the Advancement of Science).

## Data Availability

Scripts for setting up the membrane simulations in Figure 4 are available at GitHub: https://github.com/andreashlarsen/Larsen2022-review.

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
