# Peer review of "Molecular Dynamics Simulations of Curved Lipid Membranes"

_ijms, 2022, doi:10.3390/ijms23158098_

Round 1

Reviewer 1 Report

This is a well written review, the subject is timely and the topic is presented in a clear, unbiased way. It will be of use for the investigators in the area.

Author Response

See document "Answers to reviewers"

Reviewer 2 Report

In the introduction, the author writes that “curvature can target curvature-sensing lipids, peptides, and proteins to specific locations” but no references are provided to this statement. However, a lot of works have been done on this topic. See, for example, Ref. [1] and references therein.

The author should also consider the MD methods involving 1D meshless membranes for the creation of vesicles and tethers [2,3].

References

1.        Pinigin, K. V.; Kondrashov, O. V.; Jiménez-Munguía, I.; Alexandrova, V. V.; Batishchev, O. V.; Galimzyanov, T.R.; Akimov, S.A. Elastic deformations mediate interaction of the raft boundary with membrane inclusions leading to their effective lateral sorting. Sci. Rep. 2020, 10, 4087, doi:10.1038/s41598-020-61110-2.

2.        Noguchi, H. Formation of polyhedral vesicles and polygonal membrane tubes induced by banana-shaped proteins. J. Chem. Phys. 2015, 143, 243109, doi:10.1063/1.4931896.

3.        Noguchi, H. Binding of curvature-inducing proteins onto tethered vesicles. Soft Matter 2021, 17, 10469–10478, doi:10.1039/D1SM01360B.

Author Response

See document "Answers to reviewers"

Reviewer 3 Report

The need for this type of  review is clear in the lipid community. The review is very complete although in general very descriptive and with maybe too much information. The writing style for a review is not optimal, it should be significantly improved. Below some suggestions that might help:

- I do not see the point for Section 3 in a scientific review paper written by an expert

-I miss a section explaining curvature, spontaneous curvature and explicit equations on free energy contributions of bending parameters

- The author mentions in the introduction that nanoparticles and nanotubes can sense membrane curvature but no reference is added. I would suggested the following reference where nanoparticles target membrane defects:  L. Bar et al., Interactions of hydrophilic quantum dots with defect-free and defect containing supported lipid membranes, Colloids Surfaces B Biointerfaces 210, 112239 (2022)

-The section where the author compares experiments with simulations lacks explicit one to one comparison. Could the author add specific examples with direct quantitative comparisons?

Experimental section not clear at all and not direct comparison

Author Response

See document "Answers to reviewers"

Round 2

Reviewer 3 Report

The author has addressed all the questions in a satisfactory way

Author Response

I once again thank the reviewer for valuable suggestions, which have improved the manuscript.